# Deep Learning Framework with Time Series Analysis Methods for Runoff Prediction

**Zhenghe Li** [1,2,3], **Ling Kang** [1,2,*], **Liwei Zhou** [1,2] **and Modi Zhu** [3]

1   School of Civil and Hydraulic Engineering, Huazhong University of Science and Technology, Wuhan 430074, China; lizhenghe@hust.edu.cn (Z.L.); zhoulw@hust.edu.cn (L.Z.)
2   Joint International Water Security Research Center, Wuhan 430074, China
3   School of Civil and Environmental Engineering, Georgia Institute of Technology, Atlanta, GA 30038, USA; modizhu@gatech.edu
*   Correspondence: kling@hust.edu.cn

**Abstract:** Recent advances in deep learning, especially the long short-term memory (LSTM) networks, provide some useful insights on how to tackle time series prediction problems, not to mention the development of a time series model itself for prediction. Runoff forecasting is a time series prediction problem with a series of past runoff data (water level and discharge series data) as inputs and a fixed-length series of future runoff as output. Most previous work paid attention to the sufficiency of input data and the structural complexity of deep learning, while less effort has been put into the consideration of data quantity or the processing of original input data—such as time series decomposition, which can better capture the trend of runoff—or unleashing the effective potential of deep learning. Mutual information and seasonal trend decomposition are two useful time series methods in handling data quantity analysis and original data processing. Based on a former study, we proposed a deep learning model combined with time series analysis methods for daily runoff prediction in the middle Yangtze River and analyzed its feasibility and usability with frequently used counterpart models. Furthermore, this research also explored the data quality that affect the performance of the deep learning model. With the application of the time series method, we can effectively get some information about the data quality and data amount that we adopted in the deep learning model. The comparison experiment resulted in two different sites, implying that the proposed model improved the precision of runoff prediction and is much easier and more effective for practical application. In short, time series analysis methods can exert great potential of deep learning in daily runoff prediction and may unleash great potential of artificial intelligence in hydrology research.

**Keywords:** deep learning framework; runoff prediction; time series analysis; seasonal trend decomposition; mutual information; LSTM

## 1. Introduction

Runoff prediction is an important research field for hydrology, particularly where accurate runoff forecasting is critical, including flood alert [1,2], reservoir group optimal regulation [3], geological disaster prevention in karst regions [4,5], etc. Additionally, runoff exhibits a high degree of nonlinearity, ambiguity, and complexity due to the uncertainty and randomness of natural geography and human factors and the complex mechanism of runoff formation [6–8]. Therefore, building a high-precision runoff prediction model and effectively predicting a runoff time series, which are difficult tasks, have been the focus of hydrology research recently.

Since runoff prediction is a time series modelling problem [9–12], which handles the past runoff series as input and then outputs a fixed series of future runoff, the data-driven-based time series model has gained a lot of attention recently [13]. The initial attempt of the time series analysis method in hydrology was research in karst hydrology. Continuous

wavelet and discrete orthogonal multiresolution analysis methods were applied in research of rainfall–runoff relations for karstic springs [14]. Additionally, karst models with time series analysis methods were frequently calibrated and validated with time series of discharge at karst springs [15]. Nevertheless, these time series data only worked for modeling purposes if a relevant input data series were available at the same time [15,16]. For time series runoff prediction in hydrology, data quality processes, including estimating or crosschecking measured variables and filling in missing records, are a crucial part of the whole processing of datasets. There are some researchers who have paid attention to this question recently. Not only the developed time series methods themselves but also the popular deep learning networks have been demonstrated in time series discharge prediction or simulation [4,17–19]. Hence, this paper introduces the developed time series methods that have been verified in the finance field and deep learning framework into hydrology research.

Current data-driven models for time series prediction are usually built by senior researchers and require significant manual effort, including sophisticated model construction, featuring the engineering of input data and hyper-parameter tuning [8,20–22]. However, such expertise may not be broadly available, which can limit the benefits of applying data-driven models toward runoff forecasting challenges. To address this, a deep learning framework [23–26] is proposed as an effective approach that makes a data-driven model more widely accessible by automating the process of creating time series prediction models and has recently accelerated both time series research and the application of prediction models to real-world problems.

To exert the great potential of a deep learning framework in runoff forecasting problems [27], two challenges need to be overcome. First, the runoff time series data from a hydrological station often suffers from missing data and high intermittency (e.g., a high fraction of the time series data may have null values for times when a flow gauge was under repair). Second, since we aim to build a generic deep learning-based solution, the same solution needs to apply to a variety of datasets so it can capture the basic seasonal trend of runoff—both the trend of water level and discharge—as a time series.

However, few researchers have used time series analysis methods to tackle these challenges by preprocessing the runoff original data series [28,29]. Based on a former study, we designed a seasonal-trend-decomposition deep learning pipeline with upgraded original data for runoff time series prediction. In particular, the original water level and discharge time series data were upgraded by applying the mutual information method [30,31]. In addition, a seasonal trend decomposition was applied to tackle the seasonal characteristics of the runoff series. Decomposing complex time series into trend, seasonality, and residual components is an important task to facilitate time series anomaly detection and forecasting. The proposed model adopted long short-term memory (LSTM) architecture [32–39], in which a long-term memory transforms the historical information in a time series into a set of cells, and a short-term memory generates the future predictions based on these cells.

The research contributions of this paper can be summarized as follows. (1) A simple and powerful deep learning model based on time series analysis methods was proposed for daily runoff prediction. (2) The quantity of data adopted for deep learning training was analyzed and detected by mutual information. (3) The seasonal trend decomposition method can further improve the accuracy of runoff prediction. (4) Time series analysis methods combined with a deep learning framework can unleash the great potential of artificial intelligence in hydrology research.

In this article, we first analyze the main challenges of runoff time series forecasting in applying the deep learning framework and explain why we chose the time series analysis method to tackle these challenges. Then, we introduce the necessary concept for the proposed model. In the third part, we thoughtfully conduct the model structure implementation and give the data-processing details of the time series runoff data. Finally, the experimental result is concisely illustrated with the promising result.

## 2. Necessary Methods and Concepts

### 2.1. Mutual Information

Mutual information (MI) is a metric to express the amount of dependency or co-operation among variables [30,31]. It is built on the Shannon entropy of information theory [40–42]. Before introducing the details of mutual information, we first take a look at the Shannon information entropy theory. Let x be a chance variable with probabilities whose sum is 1. Then, the entropy of $x$ can be calculated by following equation:

$$H(x) = -\sum_{i=1}^{n} p_i \log p_i \tag{1}$$

Suppose two chance variables, x and y, have m and n possibilities, respectively. Let indices $i$ and $j$ range over all the m possibilities and all the n possibilities, respectively. Let $p_i$ be the probability of $i$ and $p(i,j)$ be the probability of the joint occurrence of $i$ and $j$. Denote the conditional probability of $i$ given $j$ by $p(i|j)$ and the conditional probability of $j$ given $i$ by $p(j|i)$.

Mutual information is a quantitative method based on information entropy, which can measure the relationship between two random variables that are sampled simultaneously. Unlike correlation, it does not require an assumption related to the nature of dependency, and the information it provides covers any type of linear or nonlinear relationship. In particular, it can measure how much information is communicated, on average, in one random variable about another.

For example, suppose $X$ represents the roll of a fair 6-sided die, and $Y$ represents whether the roll is even (0 if even, 1 if odd). Clearly, the value of $Y$ tells us something about the value of $X$ and vice versa. That is, these variables share mutual information.

The mutual information $I(X,Y)$ between $X$ and $Y$ is defined as

$$I(X,Y) = \sum_{i=1}^{K} \sum_{j=1}^{L} P_{XY}(x_i, y_j) \log \frac{P_{XY}(x_i, y_j)}{P_1(X = x_i)P_2(Y = y_j)} \tag{2}$$

As with other measures in information theory, the base of the logarithm in the equation is left unspecified. Indeed, $I(X,Y)$ under one base is proportional to that under another base by the change-of-base formula. Moreover, we take 0log0 to be 0. This corresponds to the limit of $x \log x$ as x goes to 0.

MI plays a crucial role in a diverse set of machine learning problems, such as independent component analysis, feature selection, and input selection, due to its attractive properties. Further information is listed in the Appendix A. It is worth noting that our new definition of mutual information has some advantages over various existing definitions. For instance, it can be easily used to do feature variables selection, as seen later. More MI properties are shown in the Appendix A.

### 2.2. Seasonal and Trend Decomposition

Additive decomposition is most appropriate if the magnitude of seasonal fluctuations or the variation around a trend-cycle does not vary with the level of the time series. When the variation in the seasonal pattern or the variation around a trend-cycle appears to be proportional to the level of the time series, then a multiplicative decomposition is more appropriate. Apparently, runoff time series data belongs to the former one, which is better decomposed additively [39].

If we assume an additive decomposition, then we can write

$$y_t = S_t + T_t + R_t \tag{3}$$

where $y_t$ is the time series data, $S_t$ is the seasonal component, $T_t$ is the trend-cycle component, and $R_t$ is the residual component, all at period $t$. Alternatively, a multiplicative decomposition would be written as

$$y_t = S_t \times T_t \times R_t \tag{4}$$

Multiplicative decompositions are common with economic time series. An alternative to using a multiplicative decomposition is to first transform the data until the variation in the series appears to be stable over time, then to use an additive decomposition. When a log transformation has been used, this is equivalent to using a multiplicative decomposition because eq 4 is equivalent to

$$\log y_t = \log S_t + \log T_t + \log R_t \tag{5}$$

Seasonal and trend decomposition using Loess (STL; where Loess is a method for estimating nonlinear relationships) is a versatile and robust method for decomposing time series [38]. The STL method can handle any type of seasonality, not only monthly and quarterly data. The seasonal component is allowed to change over time, and the rate of change can be controlled by the user, as well as the smoothness of the trend-cycle. STL can be robust to outliers (i.e., the user can specify a robust decomposition), so that occasional unusual observations will not affect the estimates of the trend-cycle and seasonal components.

The decomposition processes

Step 1. The trend item of the time series data was decomposed firstly by the method of centered moving average. When $f$ is odd, the calculation equation can be written as:

$$T_t = \frac{x_{t-(\frac{f-1}{2})} + x_{t-(\frac{f-1}{2})+1} + \ldots + x_{t+(\frac{f-1}{2})}}{f}, t \in (\frac{f+1}{2}, \ell - \frac{f-1}{2}) \tag{6}$$

When $f$ is even, the equation is written as:

$$T_t = \frac{0.5x_{t-(\frac{f}{2})} + x_{t-(\frac{f}{2})+1} + \ldots + x_{t+(\frac{f}{2})-1} + 0.5x_{t+(\frac{f}{2})}}{f}, t \in (\frac{f}{2}+1, \ell - \frac{f}{2}) \tag{7}$$

where $T_t$, $f$, and $\ell$ are the trend item, the frequency, and the sequence of time series, respectively. The calculation result is an 1D array with a time series length of 1. In order to facilitate the subsequent vector calculation, when t exceeds the domain of the above subscript, its value is a null value, such as $T_1$.

Step 2. The seasonal periodic item was removed by applying a convolution filter to the data. The average of this smoothed series for each period is the returned seasonal component.

The trend term is subtracted from the original time series, and the seasonal term $S'_t$ is obtained by averaging the values of the same frequency in each cycle. Then, the seasonal term $S'_t$ is centralized to obtain the final seasonal term. The equations are as follows:

$$S'_t = x_t - T_t \tag{8}$$

$$S_t = \sum_{i=0}^{n} \frac{S'_{t+i*f}}{f}, t \in (1, f), n = \max(n, nf \leq \ell) \tag{9}$$

Step 3. Calculate the residual item.

$$R_t = x_t - S_t - T_t \tag{10}$$

*2.3. Long Short-Term Memory (LSTM) Network*

Artificial Intelligence-based methods (including artificial neural networks and genetic algorithms) have been found to be most effective when dealing with the nonlinearity of data. However, they are lacking when it comes to handling historical dependencies in the data [27,43]. In the last few years, deep neural networks (especially recurrent neural networks, RNNs) have emerged as significant tools for dealing with nonlinearity and dependencies in data [23]. RNNs are very successful at handling short-term dependencies, but they are incapable of handling long-term dependencies due to the vanishing gradient problem.

This problem was solved with the introduction of LSTM networks. LSTM networks (improved RNNs) have been successfully used for sequence prediction and labeling tasks. The architecture of LSTM networks replaces conventional perceptron architecture with a memory cell and gates that regulate the flow of information across the network. The gating mechanism consists mainly of three gates: input, forget, and output gate. Each of the memory cells has a unit, constant error carousel, to support the short-term memory shortage for a large period of time. Figure 1 shows the structure of an LSTM memory block with one cell. In Figure 1, $c_t$ and $c_{t-1}$ denote cell states at timestamps $t$ and $t - 1$. The forget gate takes $x_t$ and $h_{t-1}$ as inputs to determine the information to be retained in $c_{t-1}$ using the sigmoid layer. The input gate $i_t$ uses $x_t$ and $h_{t-1}$ to determine the value of $c_t$. The output gate $o_t$ regulates the output of the LSTM cell on the basis of $C_t$ using both the sigmoid layer and the tanh layer. Mathematically it can be given as:

$$f_t = \sigma\left(W_f \cdot [h_{t-1}, x_t] + b_f\right) \qquad (11)$$

$$i_t = \sigma(W_i \cdot [h_{t-1}, x_t] + b_i) \qquad (12)$$

$$c'_t = \tan h(W_i \cdot [h_{t-1}, x_t] + b_c) \qquad (13)$$

$$c_t = f_t \circ c_{t-1} + i_t \circ c'_t \qquad (14)$$

$$o_t = \sigma(W_o \cdot [h_{t-1}, x_t] + b_o) \qquad (15)$$

$$h_t = o_t \circ \tanh(c_t) \qquad (16)$$

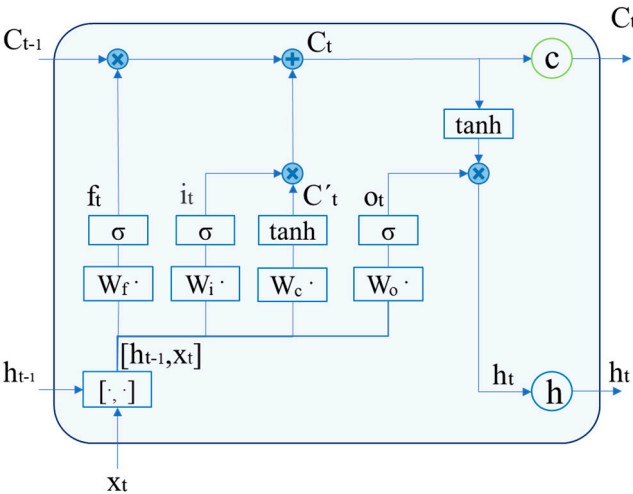

**Figure 1.** The long short-term memory (LSTM) cell.

## 3. Proposed Model

This section explains the methodology of the proposed model for runoff time series prediction and states the details of each implementing steps. A multilevel architectural diagram of the proposed MI-STL-LSTM framework is shown in Figure 2.

### 3.1. Method Impletation Steps

In the proposed model, mutual information (MI) is first adopted to calculate the sequence length of input time series and to select input variables. These are the upgraded original date series as input. Second, Seasonal and trend decomposition using Loess (STL) is used to decompose the selected series variable into three series of subcomponents. Then, the LSTM networks catch the three series of subcomponents as input to forecast the corresponding value of the next few timesteps. Finally, all the output values of the LSTM network are ensembled to produce a final prediction result for the original runoff time series.

The proposed model consists of two major phases. The first phase is the application of the time series analysis method. The original data quantity analysis and input variable selection are investigated by the application of mutual information, which reduce the modeling difficulty of the original runoff time series. The STL method is also used to facilitate time series anomaly detection and forecasting. The second phase is the prediction phase, where the LSTM architecture from the deep learning framework is adopted to tackle three subseries prediction problems and then the ensemble method is used to merge the subseries values to get final runoff values. These two phases indicate that the combination method has a higher probability of producing a forecasting model with a simple structure and better prediction ability.

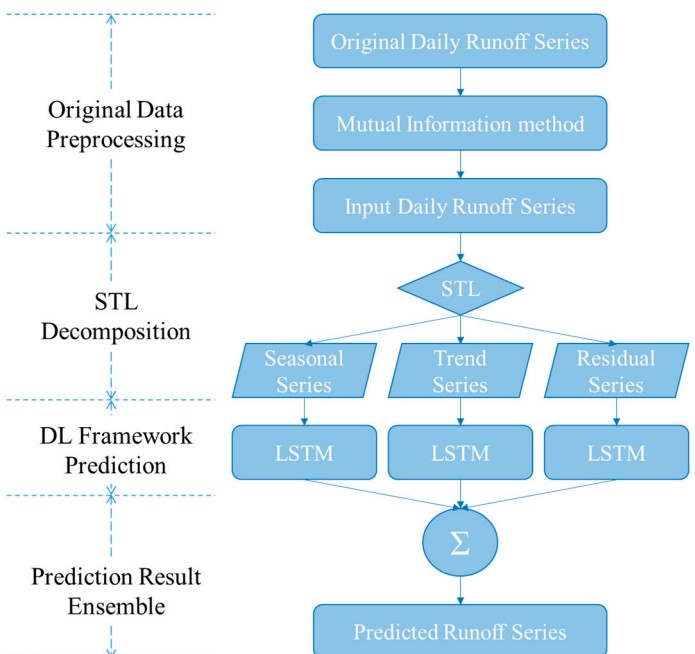

**Figure 2.** A flow chart of the proposed model, where STL for Seasonal and trend decomposition using Loess and DL for Deep Learning.

### 3.2. Model Precedure Details

The detailed procedures used by the proposed model can be roughly divided into the four steps.

Step 1: Data quantity analysis for modeling. This is a data preprocessing step that adopts the mutual information method for calculating the MI values to determine the series variables and sequence lengths for modeling. Basically, the more related series variables that are had, the higher the MI values, and vice versa. The MI values calculation of sequence length works the same way.

Step 1.1: Adopting the time series data of water level and flow variables from each station to calculate the MI values. The relation sequence of each station's variables can be ordered based on the MI values.

Step 1.2: Once the variables are determined, the input sequence length of each variable can be calculated with the MI method. This is a trial method to calculate the final result of sequence length according to a physical principle.

Step 2: Season and trend decomposition. The time series of original runoff was decomposed using seasonal-trend decomposition based on locally weighted regression to show the seasonal pattern, the temporal trend, and the residual variability.

Step 3: Prediction structure. The LSTM networks are executed independently to get the corresponding values of each subseries.

Step 3.1: With the input sequence length calculated by Step 1.2, the training dataset and testing dataset of the original runoff series can be obtained. In practice, the original input variables for each subseries should be normalized in the range of $(-1,1)$.

Step 3.2: Using the training dataset to compile and fit the LSTM model.

Step 3.3: Applying the trained LSTM to predict the target series and renormalizing the simulated output variables to obtain true predicted values.

Step 4: Ensemble final prediction result. By this time, all LSTMs get the prediction result of each subseries. Since we chose the additive method for seasonal and trend decomposition, we need to add all the subseries to ensemble the final prediction series.

## 4. Case Study

### 4.1. Study Area and Dataset Preprocessing

In this section, two national hydrological stations (Yichang and Hankou Station) located in the middle course of the Yangtze River are considered as research objects because of the importance of their runoff observations in flood prevention of the Yangtze River and in the Dongtinghu Basin.

Yichang Hydrological Station is located in the upper section of the middle course of the Yangtze River. It is the outlet station of the Three Gorges and Gezhouba Reservoirs, which are located 44 km and 6 km upstream of the Yichang section, respectively. It controls a basin area of 1,005,501 km$^2$, accounting for 55.8% of the catchment area of the Yangtze River basin [44–46]. The water level and discharge time series data of the Yichang Station can indicate the outflow parameters of reservoir group in the upper Yangtze River, which is a critical index for optimal allocation of the multireservoir system [47,48].

Hankou Hydrological Station is the first important control station for monitoring the water regime changes in the main stream of the middle reaches, after the Hanjiang River flows into the Yangtze River. It was built in 1865 and is located on the left bank of the Yangtze River, with a catchment area of about 1,488,036 km$^2$ [45,49]. The section of the basic water gauge is located in Hankou, and the flow measuring section is located about 5400 m downstream of the basic water gauge [50]. The flood control section of Hankou Station can provide the detail information for the flood control of Dongtinghu Basin, particularly in flood season.

Further information is provided in Table 1.

**Table 1.** Detail information of two hydrological stations.

| Hydrological Station | Control Basin Area (km$^2$) | Multi-Annual Average Water Level (m) | Maximum Water Level (m) | Multi-Annual Average Flow (m$^3$/s) | Maximum One-Day Flow (m$^3$/s) | The Ratio of Maximum and Minimum Flow |
|---|---|---|---|---|---|---|
| Yichang | 1,005,501 | 44.35 | 55.92 | 14,300 | 71,100 | 26 |
| Hankou | 1,488,036 | 19.19 | 29.73 | 23,800 | 76,100 | 20 |

Over the years, the two hydrological stations have been affected by upstream water, downstream water, and other comprehensive hydraulic factors. Furthermore, these factors have a different emphasis in different periods, resulting in a complex stage-discharge that is difficult to predict in time series of hydrological stations. However, it is very important to collect future water level and discharge data for flood control and water resources

management. Therefore, it is necessary to develop corresponding solutions to predict the daily runoff information of these two hydrological stations accurately and effectively for improving overall water resources management.

The daily runoff time series dataset (include water level and discharge time series datasets) of Yichang and Hankou Hydrological Stations covered the period 1995–2020. The autocorrelation test results are shown in the Appendix B. To capture more useful information, a twenty-five-year dataset (1995–2019) was used for model training. To effectively use the limited data set and avoid overfitting during training, *k*-fold cross-validation was applied; that is, the number of training set folds were set to 25. Additionally, model validation is not necessary when the fine-tuning process of a model hyper-parameter is not adopted. Moreover, to pretest the robustness of model, the penultimate year's dataset (2019) was used for validation, and the final year's dataset (2020) was utilized for testing. Figures 3 and 4 presents the time series dataset for these stations, and one can see that the daily streamflow varies within a relatively wide range, indicating the modeling difficulty of forecasting.

Data preprocessing is the first step of data analysis after setting clear objectives and defining model structure. The collected data series may have wrong values, missing values, and abnormal values, and any problematic data need to be processed first. The linear interpolation method was adopted to tackle this problem. Furthermore, to improve model efficiency and performance, preprocessing of the original input data and mapping of their attribute values between $(-1, 1)$ are necessary. Max-min normalization was used to process the datasets. The equation can be expressed as below:

$$x_n = \frac{x - x_{min}}{x_{max} - x_{min}} \tag{17}$$

where $x_{max}$ and $x_{min}$ denote the maximum and minimum values of the input data, respectively; $x$ is the observed value; and $x_n$ is the normalized value.

In this study, MI was implemented in Python separately and the other methods (seasonal and trend decomposition by Loess (STL), long short-term memory (LSTM) networks) were coded in Python language with statsmodels.tsa and keras packages.

### 4.2. Evaluation Indexes

In the following sections, four statistical indexes are used to compare the performances of the proposed forecasting models; namely, root-mean-square error (RMSE), the Nash–Sutcliffe efficiency coefficient (NSE), mean absolute error (MAE), and mean absolute percentage error (MAPE).

RMSE can effectively reflect the total error between the predicted data and observed data in all the samples, while NSE can reflect the overall deviation between the predicted data and observed data. MAE is adopted to measure the average absolute error between observed data and the simulated outputs of a forecasting model. MAPE presents the relative difference between the predicted data and observed data by a percentage value.

Basically, models with larger NSE values or smaller RMSE, MAE, and MAPE values can provide better prediction performance. The definitions of these three indexes are provided below:

$$\text{RMSE} = \sqrt{\frac{1}{n}\sum_{i=1}^{n}(y_i - \widetilde{y}_i)^2} \tag{18}$$

$$\text{NSE} = 1 - \frac{\sum_{i-1}^{n}(y_i - \widetilde{y}_i)^2}{\sum_{i-1}^{n}(y_i - \overline{y})^2} \tag{19}$$

$$\text{MAE} = \frac{1}{n}\sum_{i=1}^{n}|\widetilde{y}_i - y_i| \tag{20}$$

$$\text{MAPE} = \frac{1}{n}\sum_{i=1}^{n}\left|\frac{\widetilde{y}_i - y_i}{y_i}\right| \tag{21}$$

where $y_i$ and $\widetilde{y}_i$ represent the $i^{\text{th}}$ observed and predicted data, respectively. $n$ is the total number of data series.

### 4.3. Result and Analysis

Based on the descriptions above, the original runoff quantities and extracted subseries were modelled utilizing different methods. Primary and secondary outcome data of the individual studies are presented.

In detail, the resulting MI values indicate the suitability of the quantity of the time series data from the Yichang and Hankou Stations. From the definition of the MI value mentioned above, it can be seen that the MI value calculated by the current and former time series variables represented the shared information. When the MI value is larger, it means that the current time series variable provides more effective information for the prediction of the targeted time series variable. There was a peak MI value at 8 days for the water level variable of Yichang Station and at 5 days for Hankou Station. The procedure for the discharge variable of two hydrological stations were the same as the water level variable. Hence, the calculation result of MI values for discharge time series is presented in Appendix C.

The result shows that the runoff time series data of Yichang Station in the 8 days before were most helpful to predict the current runoff time series and that for Hankou station, it was in the 5 days before. Thus, by the application of the MI value calculation, time series input data for determining the delay days can be obtained.

Then, another time series analysis method was used to decompose the runoff time series data. The subsequence diagrams of runoff series decomposition for Yichang and Hankou Stations are shown in Figures 3 and 4. After obtaining the seasonal trend subseries and input variables, the deep learning framework—namely, an LSTM network—can be applied to runoff prediction. Consequently, Table 2 lists the detailed statistical indexes of the LSTM-based forecasting models for Yichang and Hankou station for both the training and testing phases (including the validation phases). The average percentage prediction error in the proposed LSTM network models varied from 1% to 7.35%. From the prediction results listed in Table 2, it is evident that the LSTM network model with the two time series analysis method was more reliable and accurate in prediction than naïve LSTM network model.

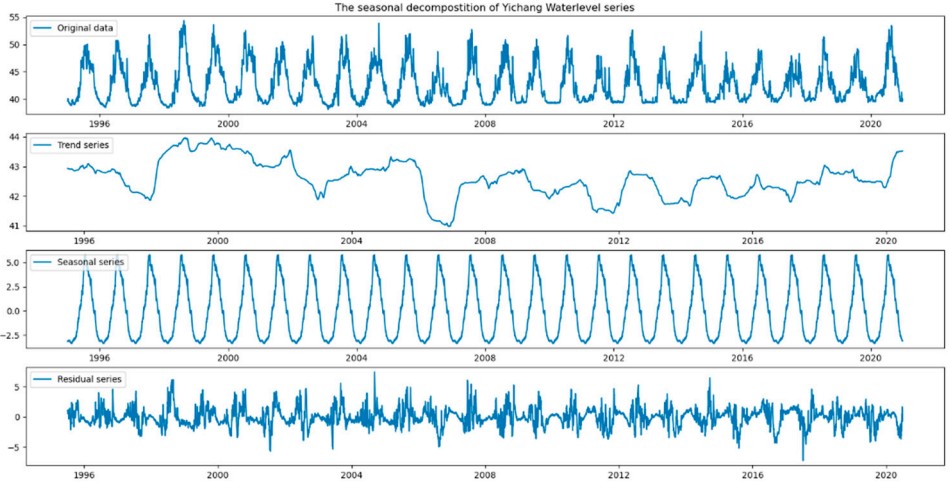

**Figure 3.** The seasonal decomposition of Yichang water level series.

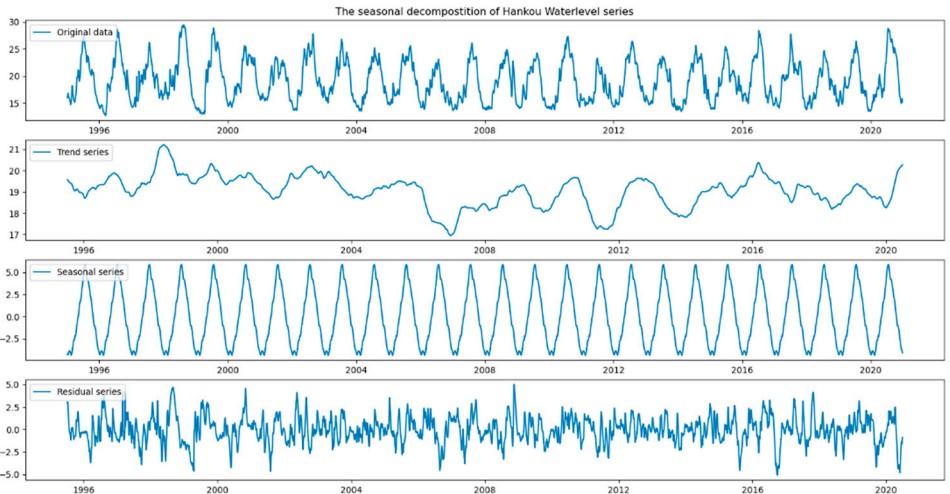

**Figure 4.** The seasonal decomposition of Hankou water level series.

Furthermore, the prediction results of water level and discharge for the compared methods for the Yichang and Hankou Hydrological Stations during the testing phase are presented in Figures 5 and 6, respectively. The *x* axis in Figures 5 and 6 represents the number of samples or data points. The *y* axis in Figure 5 indicates the water level value, the unit of which unit is meters. The *y* axis in Figure 6 is the discharge values, which is also known as runoff. The blue line represents the real observed data, while the orange represents the forecasting results of the proposed LSTM model for testing data.

In the two figures shown, almost all the predicted runoff values are perfectly fluctuating with the observed values, which indicates the promising forecasting ability of the proposed LSTM network model. In the model validation of Hankou Station, there was a small drawback; the network had a hard time predicting higher water level values. Under the influence of unexpected weather, the accuracy of model prediction will have a certain deviation.

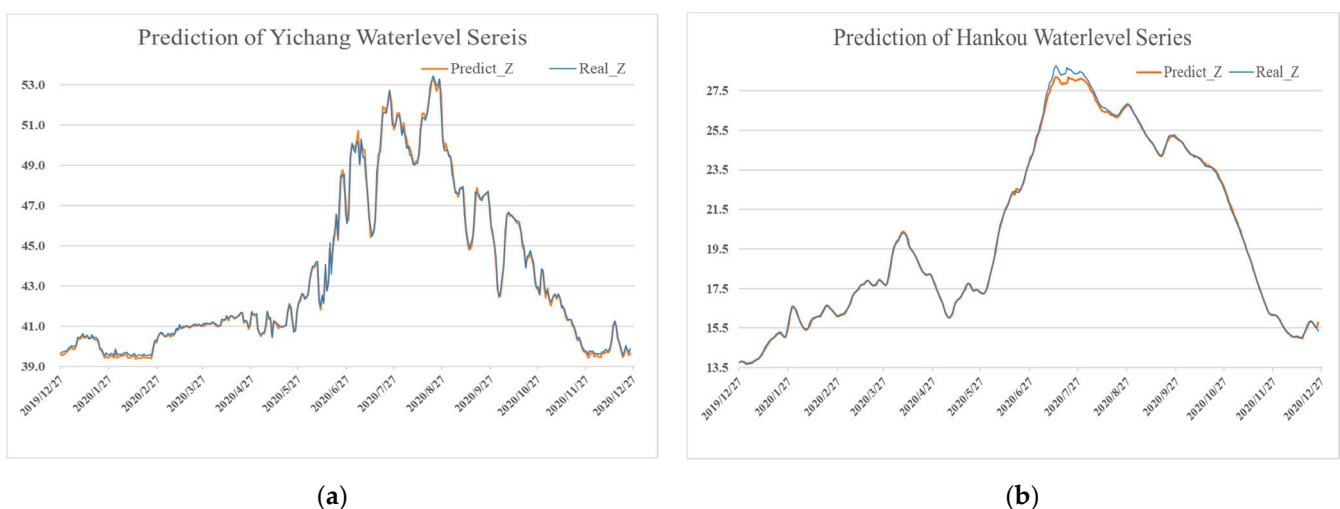

(a)                    (b)

**Figure 5.** The prediction and observation data of water level series. (**a**) The predicted result for Yichang Hydrological Station; (**b**) the predicted result for Hankou Hydrological Station.

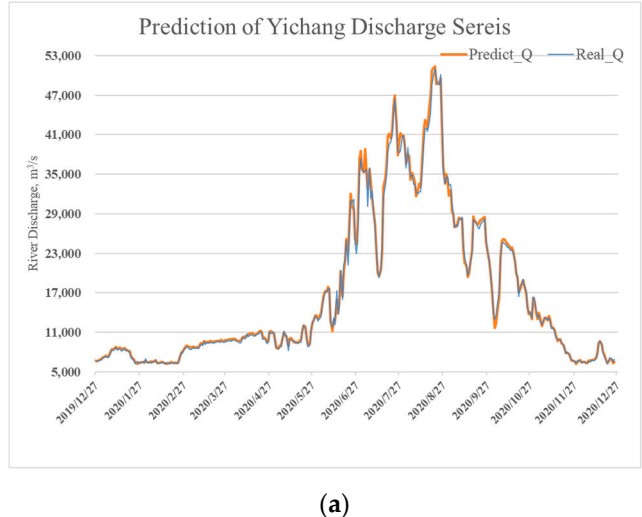
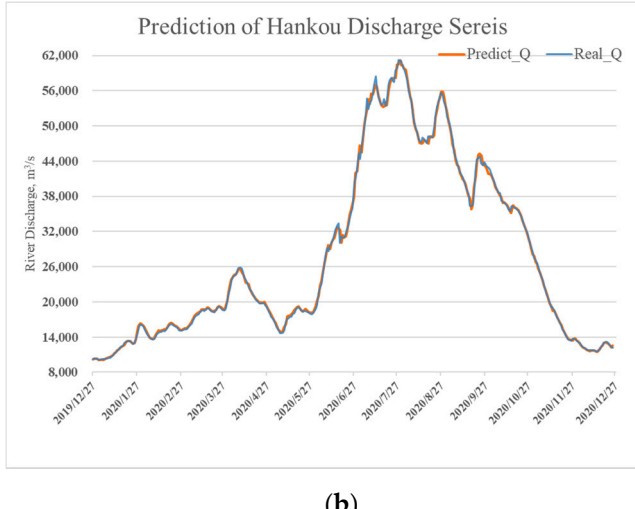

(**a**)           (**b**)

**Figure 6.** The prediction and observation data of discharge series. (**a**) The predicted result for Yichang Hydrological Station; (**b**) the predicted result for Hankou Hydrological Station.

Figures 5 and 6 present the trend diagrams of observed versus predicted runoff of the proposed LSTM model during the simulation and validation periods for Yichang and Hankou Station, respectively. From the prediction result of Yichang Station, we can conclude that the proposed model could capture the main tendency for forecasting through the application of time series analysis methods, despite the same fluctuation and trend of the water level and discharge series. For Hankou Station, it has a more complex river structure and runoff condition. The main tendency seen in Figures 5b and 6b is similar, while the proposed model can better govern the fluctuation of the discharge time series.

As shown in Table 2, the NSE values of Yichang and Hankou Stations' water level prediction was 0.966 and 0.953 during the test periods, respectively. Compared to the LSTM networks with mutual information, the proposed method exhibited an RMSE reduction of up to 60% during the testing phase. The same situation goes to the discharge time series prediction of these two hydrological stations.

There was a small deviation when STL was applied to the discharge prediction of Yichang Station from the proposed model, as can be seen in Table 2. The possible cause of this situation is that the discharge series of Yichang Station is not governed by seasonal trend but by the effect of optimal allocation of reservoir groups in the upper Yangtze River. It is worth noting that the statistical indices of prediction results and observed data were calculating by the scaler transformed data series, not the original data series. Hence, the water level result shown in Table 2 is mostly smaller than 1. For instance, these data series can be easily computed during the whole data processing process, while being governed in the same way as the original data series.

In short, it can be concluded that the proposed LSTM networks model with two time series analysis methods is an effective tool for forecasting the daily runoff series of a hydrological station.

**Table 2.** The results of different forecasting methods for the Yichang and Hankou Hydrological Stations.

| Hydrological Station | Time Series Variable | Models | Nash–Sutcliffe Efficiency Coefficient (NSE) | Root-Mean-Square Error (RMSE) | Mean Absolute Error (MAE) | Mean Absolute Percentage Error (MAPE) (%) |
|---|---|---|---|---|---|---|
| Yichang | Discharge | Naive LSTM | 0.8754 | 1080.7725 | 822.8122 | 6.0155 |
| | | LSTM with mutual information (MI) | 0.8926 | 527.6695 | 272.0359 | 2.8931 |
| | | Proposed model | 0.9694 | 634.7099 | 380.9427 | 2.0967 |
| | Water level | Naive LSTM | 0.8212 | 0.4278 | 0.3121 | 6.904 |
| | | LSTM with MI | 0.8666 | 0.3966 | 0.2818 | 6.203 |
| | | Proposed model | 0.9661 | 0.1529 | 0.1098 | 2.461 |
| Hankou | Discharge | Naive LSTM | 0.8184 | 1078.5187 | 838.7032 | 7.3562 |
| | | LSTM with MI | 0.8620 | 699.0840 | 364.2954 | 5.3965 |
| | | Proposed model | 0.9711 | 450.5172 | 281.5328 | 0.958 |
| | Water level | Naive LSTM | 0.8258 | 0.4918 | 0.3366 | 6.039 |
| | | LSTM with MI | 0.9066 | 0.4386 | 0.3173 | 5.626 |
| | | Proposed model | 0.9530 | 0.1348 | 0.0774 | 3.396 |

## 5. Conclusions

LSTM networks have been popularly applied to hydrology research because of their powerful predictive capacity. However, little is known about the input data quantities that are needed for data-driven model implementation or about an LSTM network's robustness when used under seasonal change beyond a calibration period. In this work, we combined the deep learning framework, an LSTM network, and two time series analysis methods under two different hydrological stations in the middle course of the Yangtze River. The main conclusions are summarized as follows. In particular, water level and discharge time series variables were used to demonstrate the forecasting ability of the proposed model.

The model's demand for a large amount of data seems to be the major limitation hindering the practical application of an LSTM network. However, the quantity of data needed for such a data-driven model have not been deeply investigated. On the other hand, the seasonal characteristics of runoff also did not play its due role in runoff prediction. Time series analysis methods have performed well and gained a lot of importance due to their effectiveness in handling time series problems. By applying the mutual information method, the calculated MI value gave exact information about input data quantities and selected input variables. With seasonal and trend decomposition, the proposed model showed the robustness of capturing the seasonal trend for runoff.

There are several advantages of using the proposed approach for runoff prediction as compared to other existing approaches. (1) The proposed framework can effectively handle nonlinear complexities and short-term and long-term dependencies of the runoff time series data. (2) The proposed model is completely adaptive for data quantity analysis and provides support for capturing the seasonal trend of runoff series data. (3) The simulation results indicate that the proposed LSTM network model has minimum prediction errors. (4) The developed framework can be easily generalized to estimate runoff for other hydrological station as it is purely dependent on historical data only.

## 6. Discussion

For future work, various nonlinear exogenous features such as climate conditions and economic variables can be investigated for trend analysis of runoff patterns. Further, various optimization techniques can be designed to improve the prediction accuracy of learning models. Additionally, as the open-source code of the deep learning community becomes more accessible, we also ask for more joint research and collaboration among peer researcher to demystify the use of artificial intelligence for hydrology research.

Finally, there are three potential directions for future research based on the proposed model. The first direction is model stochasticity. Even though this study reports an effective model for runoff prediction and presents promising predicted results, it did not validate the stochasticity of the result [51]. The second potential direction is to introduce an effective and simple time series analysis method for solving the time series problems. Additionally, future research needs to pay attention to the validation of a combination model, the plausible result of which did not equal a valuable component. Third, is creating a benchmark for runoff prediction [37,52]. As the deep learning framework for runoff prediction with a data-driven model gets more and more attention, research for a benchmark in the near future for use as a comparison basis is also very important.

**Author Contributions:** Z.L. proposed the innovation and conducted the programming. Z.L. wrote the draft of the paper. L.K. and L.Z. revised the original draft and corresponded regarding the comments from reviewers. M.Z. provided necessary code and program corrections. All authors have read and agreed to the published version of the manuscript.

**Funding:** This research was financially supported by the National Key R&D Program of China, grant number 2016YFC0402202.

**Data Availability Statement:** More data shall be found at https://github.com/teetyler/STL-LSTM.

**Acknowledgments:** The authors gratefully acknowledge discussions with Feifei He, Kuaile Feng and valuable individuals that contributed to the development of this paper.

**Conflicts of Interest:** The authors declare no conflict of interest.

## Appendix A

MI as a time series analysis method that plays a crucial role in a diverse set of deep learning problems [30], such as independent component analysis, feature selection, and input selection, due to its attractive properties listed below:

- MI is nonnegative, $I(X, Y) \geq 0$.
- MI has a symmetricity property, $I(X, Y) = I(Y, X)$.
- MI can handle detecting any type of dependencies, even with no correlation.
- MI has an invariance property for linear transformations of X and Y.
- MI has an association with entropy, which is a quantity of uncertainty that a random variable carry, $I(X, Y) = H(X) + H(Y) - H(X, Y) = H(X) - H(X \mid Y) = H(Y) - H(Y \mid X)$, where $H(.)$ represents entropy.
- $I(X, Y) = 0$, if and only if the components X and Y are statistically independent.

## Appendix B

Generally, the stationariness of an input variables series has a direct effect on the final forecasting results. In this study, autocorrelation was selected as a potential indicator for identifying the stationarities of input variables.

It is clear that the two hydrological station runoff series are stationary from the autocorrelation subplot in the bottom right of Figure A1. Hence, the following prediction can flow with observed data.

**Appendix C**

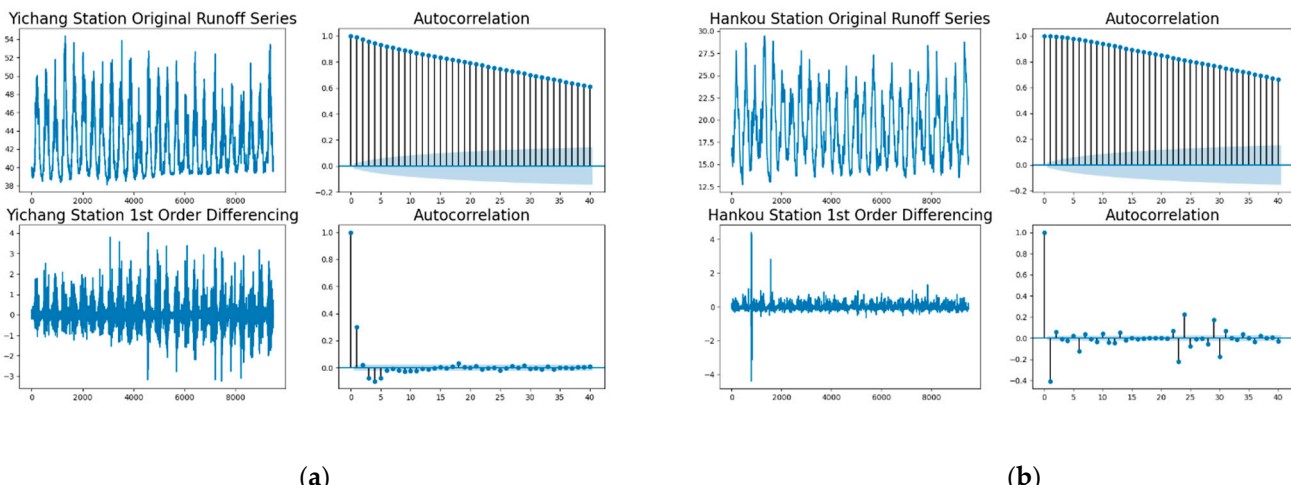

(**a**)                                                              (**b**)

**Figure A1.** The autocorrelation of two hydrological station runoff series. (**a**) The Yichang Station original runoff series and the first-order difference series with their autocorrelation values. (**b**) The Hankou Station original runoff series and the first-order difference series with their autocorrelation values.

From the definition of the MI value mentioned in Appendix A, it can be seen that the MI value calculated by the current and former time series variables represents shared information. Additionally, with the properties of MI value, when the MI value is larger, it means that the current time series variable provides more effective information for the prediction of the targeted time series variable. With the help of computer simulation and computation, there is a peak MI value at 7 days for discharge variable of Yichang Station and at 6 days for Hankou Station.

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
