# Peer review of "Deep Learning Framework with Time Series Analysis Methods for Runoff Prediction"

_water, doi:10.3390/w13040575_

Round 1

Reviewer 1 Report

The theme of this manuscript is quite interesting and I think many other researchers are applying the similar methods on other time series data sets for the prediction.
The runoff time series data sets were analyzed in this research and there were only two cases for the verification of the results with observed data.
I hope the author could add several more cases to see the various accuracy levels of the results and analyze errors in discussion section for future studies. 

Author Response

We only choose the water level as the experiment time series dataset for testing the proposed model because of the uniqueness of water level, where there is a specific functional relationship between water level and discharge variable. As we did not arrange any experiment about discharge time series dataset, there is a drawback for demonstrating the potential of Deep Learning Framework and Time Series Analysis methods.

By taking the suggestion from reviewer, we made the whole test experiment of discharge time series variable. Surprisingly, we found that the result of discharge prediction is better than the water level prediction by giving the same situation for both Hydrological Station. The detail revision can be seen at Line#64 (specifical announcement), Line#246 (case study), Line#358 (result analysis) and Line#389(conclusion). Please open the "Track Changes" function in Microsoft Word for more revision information.

For more information, Please check the enclosed cover letter. Thanks and May you have a wonderful vacation.

Reviewer 2 Report

The paper is well written, the conclusions are supported by the analysis of the data and therefore the paper my be interesting for publication after considering few but important comments below:

  1. Introduction --"Runoff prediction is an important research field for hydrology.." It is correct but it is not correctly included in paper. The authors used only water level time series and they avoid the most important hydrological characteristics--time series of the discharge data. 
  2. Case study--the authors write in line 250: The daily runoff time series data set...covered period 1995-2000...and the graph of water level is presented
  3. Figure 4, Figure 5 present water level series

The main suggestion is that the time series of the discharge data must be included in analysis as the main hydrological series. 

Literature also may be extended with some recent citations connected with time series in hydrology and karst hydrology. 

Author Response

We have made the whole test experiment of discharge time series variable. Surprisingly, we found that the result of discharge prediction is better than the water level prediction by giving the same situation for both Hydrological Station.

As the reviewer#2 pointed out, we also added some recent research reference about the influence of runoff prediction for Karst Hydrology. Since the hydrological principles in the Karst region are more complicated, the runoff prediction is essential for flood and drought alert.

For more information, Please check the enclosed cover letter. Thanks again and May you have a wonderful vacation.

Round 2

Reviewer 1 Report

Overall efforts to revise the original manuscript and some improvements were checked.
If the authors showed several other time series data sets and results from other watersheds then the methods could have been proved to be strongly recommended for the prediction of runoff or water level data.

Author Response

Dear reviewer,

Thanks for your graceful review notes.

With the kindly suggestion you have proposed in the review notes, we have added extra simulation experiment. The added experiment also shown the promising result. We believe that the prediction result of two important National Hydrological Station already demonstrate the forecasting potential of proposed model as presented in the revised article. For more case studies of time series datasets, we will try to get access to datasets and collect more valueable data.

Thanks again and wish you all the best.

Reviewer 2 Report

The literature review must describe a systematic way of colllecting previous research and can help to provide an overview of the area of investigation so I suggest the extension of the literature with papers linked with the application of time series analysis in karst hydrology (Mangin, Labat, Hartmann ...). I recommend also the paper: Literature review as a research methodology: An overview and guidelines by Hannah Snyder (2019) as the base for this suggestion. 

Author Response

Deer reviewer,

Thanks for your graceful review notes.

With the kindly suggestion you have proposed in the review notes, we have added a paragraph to describe a scientific way of time series research and provide an overview of the area of discharge investigation. The detailed information can be found in the Line#44-57. The authors, provided in the review notes, who have paid tremendous attention to improve the time series analysis methods and model the rainfall-discharge relation in karst region or hydrological process.

These research articles or reviews are very helpful for knowing the basic but crucial problem which cannot fix by the exist framework. And with the thoroughly investigation of time series discharge research, not just for karst hydrology, the research direction of runoff prediction can be easily defined.

With the development of Artificial Intelligence, many research frameworks proposed and directly applied without caring the professional situation. Giving all the consideration, we improved the data preprocessing part with the application of time series analysis methods and combined the advantages of Deep learning framework to fixed the long-term dependency of series data. Our scientific research is also a small step to promote the simulation and prediction of hydrological processes in the middle Yangtze River by combining Deep Learning framework and time series analysis methods.

Thanks again and wish you all the best.

Zhenghe